# Challenges and Opportunities of Deep Learning for Cough-Based COVID-19 Diagnosis: A Scoping Review

**DOI:** 10.3390/diagnostics12092142

**Published:** 2022-09-02

**Authors:** Syrine Ghrabli, Mohamed Elgendi, Carlo Menon

**Affiliations:** 1Biomedical and Mobile Health Technology Lab, ETH Zurich, 8008 Zurich, Switzerland; 2Department of Physics, ETH Zurich, 8093 Zurich, Switzerland

**Keywords:** cough audio signals, COVID-19, neural networks, deep learning

## Abstract

In the past two years, medical researchers and data scientists worldwide have focused their efforts on containing the pandemic of coronavirus disease 2019 (COVID-19). Deep learning models have been proven to be capable of efficient medical diagnosis and prognosis in cancer, common lung diseases, and COVID-19. On the other hand, artificial neural networks have demonstrated their potential in pattern recognition and classification in various domains, including healthcare. This literature review aims to report the state of research on developing neural network models to diagnose COVID-19 from cough sounds to create a cost-efficient and accessible testing tool in the fight against the pandemic. A total of 35 papers were included in this review following a screening of the 161 outputs of the literature search. We extracted information from articles on data resources, model structures, and evaluation metrics and then explored the scope of experimental studies and methodologies and analyzed their outcomes and limitations. We found that cough is a biomarker, and its associated information can determine an individual’s health status. Convolutional neural networks were predominantly used, suggesting they are particularly suitable for feature extraction and classification. The reported accuracy values ranged from 73.1% to 98.5%. Moreover, the dataset sizes ranged from 16 to over 30,000 cough audio samples. Although deep learning is a promising prospect in identifying COVID-19, we identified a gap in the literature on research conducted over large and diversified data sets.

## 1. Introduction

The coronavirus disease 2019 (COVID-19) is an infectious disease caused by the severe acute respiratory syndrome coronavirus-2 (SARS-CoV-2). At the time of writing, the World Health Organisation (WHO) has reported more than 510 million confirmed cases, including more than 6 million deaths [1]. Two years after the coronavirus outbreak, the virus continues to spread around the world. Therefore, prompt and precise identification of the virus is indispensable. Various testing methods are available to detect the hallmarks of the virus. Among them, reverse transcription-polymerase chain reaction (RT-PCR) testing is currently considered the gold standard. However, none of the existing tests are 100% accurate [2]. Moreover, RT-PCR tests can be time-consuming, unaffordable, unavailable at times or in some areas, and pose a risk of transmitting the virus to healthcare workers and others. In light of the foregoing, fast and accessible prescreening tools are urgently needed to limit the spread of COVID-19.

To this end, there have been increased efforts to construct predictive diagnostic systems using a wide range of media, including mechano-acoustic signatures of vital signs such as heart rate, respiratory rate, temperature, and other respiratory biomarkers such as breath, speech, or cough. Notably, the field of healthcare is undergoing a digital health revolution, in which artificial intelligence (AI), big data, mobile devices, and other technological innovations are giving rise to new horizons. Machine learning techniques, such as artificial neural networks, are already widely used for medical applications.

Modeled on the propagation of biological neurons, a neural network is a sequence of node layers. Given enough data about a feature *x* and the corresponding data label *y*, neural networks succeed remarkably in solving mapping from *x* to *y*. Through learning by example, neural networks can detect diseases without stipulating a specific way to identify them, making them singularly valuable. In our survey, we examine how cough sounds have been utilized for the training, evaluation, and validation of neural network algorithms aimed at helping healthcare workers detect the coronavirus.

Cystic fibrosis, pulmonary edema, pneumonia, chronic obstructive pulmonary disease, and bronchitis are all respiratory diseases with a common symptom: cough. Notably, they have been the center of deep learning multi-classification algorithm development. Research in the last few decades has shown outstanding performance results (reaching higher than 90% in accuracy [3] and sensitivity values [4,5,6]).

When a cough is produced, it is generally composed of three phases [7]: inhalation, exhalation against a closed glottis, and release of air from the lungs following the opening of the glottis [8]. There are two types of coughs: wet, mucus-productive cough and dry (nonproductive) cough. These have been repeatedly analyzed and characterized by phase duration or frequency for classification tasks [9].

Figure 1 shows insights into the amplitude shapes of the raw data of two distinct coughs. Coughs from COVID-19-positive and COVID-19-negative patients can appear different; however, no information or interpretation can be read from this single graph, and only statistical analysis is relevant. For instance, Pahar et al. [10] statistically found that COVID-19 coughs are 15–20% shorter in duration than non-COVID-19 coughs. Singh et al. [11] stated that for involuntary healthy coughs, the energy distribution has values in all frequencies, and two to three harmonics are visible; for other coughs, there is no clear pattern in the structure of harmonics, and the second phase has two to three harmonics below one kilohertz.

Despotovic et al. [13] listed the 10 most informative acoustic patterns in cough and breath sound among hundreds of extracted attributes with the help of a mutual information criterion. They later identified the most relevant features in cough signals that can detect COVID-19, such as spectral harmonicity, root mean square energy, and spectral slope. They explained that the energy spectrum of the cough of a COVID-19 patient shows low frequencies at the beginning and a shift in frequency values later, perhaps due to the pain and effort required for the patient to cough.

In sum, the total or phase duration, peak frequency, location, intensity, pattern, energy spectrum, and power ratio can help distinguish between healthy and COVID-19-positive patients.

## 2. Methods

### 2.1. Study Guidelines

This review was conducted according to the Preferred Reporting Items for Systematic Reviews and Meta-Analyses statement (PRISMA) [14]. A review protocol was drafted using the Preferred Reporting Items for Systematic Reviews and Meta-Analyses Protocols [15] for internal use amongst the research team but it was not externally published or registered prospectively.

### 2.2. Search Strategy and Study Eligibility

The focus of the search had to be limited to cough and neural network uses for the classification of COVID-19 using cough signals. To find pertinent research on this topic, the following search terms were employed verbatim: cough, COVID-19/coronavirus/SARS-CoV, detection/classification/predict/diagnosis/testing/screening, and neural/deep network/learning. Other extensively used abbreviations, such as convolutional neural network (CNN), recurrent neural network (RNN), and deep neural network (DNN), were also utilized. To ensure the thoroughness of the review, PubMed, Embase, Science Direct, Google Scholar, IEEE, and ResearchGate were searched for papers published between 1 January 2012 and 1 January 2022. Gray literature was not included in this review in an attempt to only include peer-reviewed studies. This timeframe was chosen to reflect advances in smart sensors, artificial intelligence technologies, and their kiosk applications in medicine. The search for this review was completed in April 2022.

### 2.3. Inclusion and Exclusion Criteria

In the first pre-screening phase, duplicate records and papers published in languages other than English or that were not accessible were discarded. The screening phase consisted of several steps. First, all the reviews, case reports, or corrigenda papers were removed. Then, all the studies considered irrelevant based on their title were discarded. Finally, the abstracts of the papers were reviewed, and the key information was extracted from the main text. The criteria for screening were as follows:papers using neural network methods to diagnose COVID-19 on biomedical imaging;papers using cough audio signals to diagnose COVID-19 through classical methods of machine learning classification; andpapers using cough to diagnose COVID-19, not as an audio signal but as a symptom feature, were filtered out.

Once all the papers were selected and grouped, the last step was to go through the results again, ensuring that nothing was omitted erroneously.

Although the use of cough and neural networks for the classification is a requisite and paramount criterion, papers including breathing or other acoustic signals along with coughs, as well as papers that include shallow or “classical” classification machine learning algorithms along with neural networks, were still considered in the screening. Similarly, articles that aimed to perform multi-class classifications of COVID-19-positive vs. healthy individuals and other illness classes were used in the synthesis. One reviewer (SG) conducted the literature search, and two reviewers (SG and ME) independently screened the titles, abstracts, and full texts for potentially eligible studies. Reference lists of eligible studies were also hand-searched, but no additional studies were included.

### 2.4. Data Extraction and Risk of Bias

One author (SG) conducted the literature search, and two authors (SG and ME) independently screened the titles and abstracts for potentially eligible studies. Each potential study for inclusion underwent full-text screening and was assessed to extract study-specific information and data. For each of the included articles, the information gathered was the title of the model(s) used for the classification, the models’ performances, a rough description of the data, and extracted features without all the details. However, the number of layers, type of layers, loss of function, and others used to train or validate a model, the pre-processing of the data, or the implementation steps taken within the papers were not reported. If other tasks were executed during the implementation steps aside from the diagnosis of COVID-19 from cough and if several attempts at neural network model tuning were described, these were also not present. Finally, in retrieving the performance metric results, some evaluation methods, such as unweighted average recall (UAR), Kappa coefficient, and F1-score, were scarcely used and, thus, were not reported.

## 3. Results

### 3.1. Study Selection

In this section, we present our findings regarding the selection of studies. The flow diagram in Figure 2 visually summarizes the screening process. Initially using broad terms and then moving to more specific keywords, the searches generated more than 200 results. Several criteria were applied during the screening to gather the literature for review.

Out of hundreds of search results, 33 scientific articles were reviewed, as shown in Table 1. These articles differed in the scope of their studies. Some articles aimed to compare classical machine learning vs. deep learning models (e.g., support vector machine (SVM) vs. DNN), distinct types of deep learning models (e.g., CNN vs. RNN), distinct architectures of one type of model (e.g., CNN vs. residual neural networks (ResNet)), or hyperparameter tweaking of a single model. Numerous articles examined the impact of feature extraction on the performance of their model(s). Other articles compared the modalities of cough, breath, and voice audio signals to assess the performance of their model(s).

The dataset sizes ranged from 16 to over 30,000 samples. Among the most frequently used datasets were open-source Coswara [16], Virufy [17] and Coughvid [12]. Each dataset has attributes and metadata, usually collated from users’ survey information via web platforms or smartphone apps. For instance, some datasets included supplementary metadata and features, such as symptoms, smoking habits, and other respiratory illnesses for each audio sample. While some of these were used as additional features for training, others were applied as labels for different classification tasks (e.g., symptomatic COVID-19-negative vs. COVID-19-positive). These features created an opportunity to use metadata for further investigations and more closely reflected the reality during the training and validation of the model. In the vast majority of cases, the datasets were imbalanced: the pool of COVID-19-negative samples was considerably larger than that of the COVID-19-positive samples.

#### Process Description

The flowchart in Figure 3 illustrates the simplified pipeline of procedures followed by the studies that were examined during the survey. The first step was gathering and defining the classification task data. Besides a few instances where the extraction of cough sounds was sourced from hospital patients or publicly available videos (interviews on YouTube [18] of COVID-19 patients), the datasets of the cough sounds were mostly crowdsourced via website platforms or smartphone apps. The most common bit rate was 24 bits; the sampling rates were 44.1 and 48 kHz, which can be altered and down-sampled. The main pre-processing steps were bandpass filtering, noise reduction, and other data filtering figures. After the selection and pre-processing of the data, feature extraction was undertaken. In addition to raw audio signals, spectrogram images and vectorial features can be fed into neural networks. For instance, spectrograms are extensively used and considered sound fingerprints in audio recognition. Music identification apps, such as Shazam, are leveraged in this context. The mel spectrogram is non-linear, with frequencies converted to the mel scale. The mel scale is a logarithmic scale of frequencies judged as equal in distance from one another.

The mel-frequency cepstrum (MFC) is a power spectrum based on frequencies in the mel scale. Cepstrum computation is a tool to study periodic structures in frequencies. Since they are an inverse Fourier transform, MFCs are the coefficients of amplitude that concisely describe the overall shape of a spectral envelope [19]. Audio cepstral coefficients include features such as gammatone cepstral coefficients, log-energy, delta, and delta-delta coefficients. Gammatone cepstral coefficients are defined as a biologically inspired modification of the MFCCs using gammastone filters, the log-energy measures the sum of energy of each frame of a signal in the logarithmic scale, the coefficients delta and delta-delta, are the change in gammastone cepstral coefficients and the change in delta values, respectively [20]. The latter coefficients, spectrograms, scalograms, and extracted features are either concatenated or passed onto some method for feature selection (e.g., cross-validation). The results are intended to be fed into the appropriate model for training.

During this preliminary phase, the Visual Geometry Group (VGG) [21], a CNN that is well-known for image classification, is the primary neural network architecture utilized for feature extraction purposes, but not as a model for training or classification [22]. There are not enough results to effectively conclude that without feature extraction, the performance of a deep learning strategy is consequently attenuated. However, deep learning architecture can combine feature extraction with classification. Indeed, it has been used with or without deep learning for classification. For this reason, one can ask, “Why extract features at all?”. However, few articles compared the implementation and combination of features for a fixed model. Performing sequential forward selection to select 13 features led to an improvement in the area under the receiver operating characteristic curve (AUC) from 77.9% to 93.8% [10]. Although all features are not the main focus of our study, their presence is essential to understanding the full scope of the task. For any machine learning exercise, the choice of a model depends on the data, data size, and data representation.

### 3.2. Models

#### Definition of Models

Not all neural network architectures are created equally.

**CNN**: CNN is notably powerful for image classification tasks, as the execution of dimensionality reduction suits many parameters in an image. CNN takes four-dimensional inputs (a two-dimensional image, depth, and batch size) to which convolution operations are applied, outputting a feature map. The latter is then flattened (1D) to fully connected layers.

**ResNet**: ResNets are a type of CNN that introduces skip layers to avoid the vanishing gradient problem; the layers are gradually restored as the feature space is learned. Among the neural network architectures, especially for visual object classifications, such as AlexNet, DenseNet, Inception, and EfficientNet, ResNet is the most predominant among the models encountered during this study, particularly ResNet-50.

**RNN**: While CNNs use convolution layers to filter data, RNNs reuse anterior activation functions from other nodes to produce the next output in a sequence. They are designed for temporal and sequential input and are widely used for speech recognition.

**Long Short-Term Memory (LSTM)**: One prevalent example of RNN is the LSTM algorithm, which is equipped with specific gates, such as the forget gate, to remedy the vanishing gradient problem with RNNs.

**Transfer Learning**: When only a small dataset is at hand, it is generally useful to avail oneself of models that are pretrained on large datasets, from freezing the weights of the first input layers of the pretrained model—training the softmax function with the small dataset—to re-initializing the entirety of the pretrained weights.

In the context of transfer learning, Pahar et al. [23] utilized a total of 11,202 cough sounds from the datasets TASK, Brooklyn, Wallacedene, and Google Audio Set &Freesound. Dentamaro et al. [24] made use of an open-source dataset, UrbanSound 8K, which is comprises 10 classes of street sounds and no cough [25]. With transfer learning, their model’s AUC improved by approximately 9%.

Based on the histogram in Figure 4, we can review the models that have been used for COVID-19 classification and their counts. We can see that CNNs are a large majority. The count of CNN usage comprises the instances where CNNs were used in research regardless of their shape but does not include known architectures cited within articles under the CNN category, such as ResNet-50. The latter is also a popular architecture among the studies surveyed, but it is not the only “borrowed” architecture implemented, In fact, Loey and Mirjalili [26] juxtaposed ResNet-50 with ResNet-18, ResNet-101, GoogleNet, NASNet, and MobileNet V2. In two articles, CNN and RNN architectures were used and compared. LSTM showed higher overall results than ResNet-50 in the first study [10], while the LTSM showed lower results overall than CNN and ResNet-50 in the second [23].

The popularity of CNN models can be explained by the fact that sound recognition can be translated into a visual recognition task for which CNN has demonstrated great success. Amoh and Odame [27] detailed two methods in the context of a cough detection task, namely, CNN for visual recognition and RNN for sequence-to-sequence labeling, reporting in their performance results that CNN yields a higher specificity than RNN. Pahar et al. [10] broached the discussion of the evaluation of shallow, or “classical,” machine learning classifiers, namely, logistic regression (LR), k-nearest neighbor (KNN), and SVM, against deep neural networks: multilayer perceptron (MLP), CNN, LSTM, and ResNet-50-based neural network architecture. Notably, LR, KNN, and SVM reached AUC values of 73.6%, 78.1%, and 81.5%, respectively, whereas the neural network models reached 89.7%, 95.3%, 94.2%, and 97.6% for MLP, CNN, LSTM, and ResNet-50, respectively.

For the evaluation and validation of the performances of the models, a few strategies have been established. As a baseline, Soltanian and Borna [28] used classical machine learning (SVM, random forest, KNN), Coppock et al. [29] used a classical SVM binary classifier without pre-processing, and Andreu-Perez et al. [30] applied Auto-ML. Meanwhile, Chaudhari et al. [31] presented a multi-branch ensemble architecture of ResNet-50, a simple feedforward network, and DNN against the ResNet-50 model performance. Soltanian and Borna [28] showed that quadratic-based CNNs could provide higher accuracy when compared to “ordinary” CNN. Often, studies used their models to compete against the existing ones in the literature [26,32].

### 3.3. Existent Bias Reported

The gender of a patient’s cough is perceptible to the human ear. [33] For machines, this difference is also evident in the data. Moreover, when the performance of one neural network model is assessed for several separate subgroups, differences arise. Han et al. [34] thoroughly detailed the performance of a CNN for distinct subgroups of gender, age, symptom manifestation, medical history, and smoking status. When separated, the subgroups yielded higher performance (the difference was approximately 10% on average). Against a control group of nonsmokers, the AUC performance of the model trained on smoker subgroups was roughly lower. Similarly, against a control group of patients with no medical history, the AUC performance of the model trained on the subgroup of patients suffering from respiratory ailments was also lower. Han et al. [34] concluded that the fluctuations in a model’s performance are due to the volume of the subgroups, that medical history cannot confuse their model, and that the model generally performs better in predicting symptomatic COVID-19.

In the study of Chowdhurry et al. [35], the values of relative closeness scores were calculated, with an average 54.0% for the asymptomatic category of cough samples and 67.4% for the symptomatic category. These results indicate that their model is more accurate at predicting symptomatic or greater infections of COVID-19. Imran et al. [36] validated their architecture’s ability to predict COVID-19 from two diseases (i.e., bronchitis and pertussis). Additionally, Pahar et al. [23] placed side by side their specificity, sensitivity, accuracy, and AUC computations for ResNet-50, LTSM, CNN, MLP, SVM, and KNN for each of the datasets from Coswara, Sarcos, and ComParE. The datasets differed in their sampling rates, labels, and subject sizes. For example, the dataset Sarcos was used as a validation set for classifiers trained on the Coswara data, and an AUC of 95.4% was attained. The Sarcos dataset’s sampling rate was 44.1 kHz, but it contained 26 COVID-19-negative and 18 COVID-19-positive audio samples. Finally, Han et al. [37] collated separate cough, voice, and breath segment model performances with the fusion of all respiratory data using VGGish, the pretrained CNN from Google. The best performances recorded were for the three modalities combined—AUC, sensitivity, and specificity—with a difference of about 4–6%.

## 4. Discussion

The evaluation of a model consists of computing a model’s predictions for a specific dataset using different evaluation metrics, as shown in Table 1. However, does a high evaluation metric score indicate that a model is better than one with a lower score, or is this due to statistical bias or improper metric calculation? In the context of the research surveyed, the datasets used for the evaluation step were too small or did not provide enough information for us to form an opinion on their validity. Additionally, at the time of writing, the datasets used in the encountered articles did not span a period long enough to include all COVID-19 variants. Thus, in appraising a machine learning model, one needs to understand its dependency on data, including the date of a paper’s publication, the data collection interval, and the diversity and ethnicity representation in the cough sounds.

A straightforward yet in-depth comparison of the neural network models is challenging to formulate. They generally lacked subject counts from different ethnicities, ailments, and smoker statuses. While some articles elucidated all the steps to describe the number of layers experimented with, others denoted using a simple CNN or RNN. Likewise, some articles referenced the name of the architecture used, whereas others detailed how it was incorporated. While popular CNNs have proven their efficacy, they also entail the use of fully connected layers, the added complexity of which makes them prone to overfit data. Although we noticed that some strategies had been used to overcome this, such as L2-regularization, overfitting remains a challenge but could not be addressed in this survey.

To assess a model’s performance, one needs to choose the appropriate evaluation metrics and the data to apply them. Although it constitutes the basis of the paper’s results, the assessment of model performance ought to be taken with a grain of salt by the reader. Some immediate limitations are as follows:(1)having just one metric presented in a paper provides us with one understanding of performance and leads to the impossibility of establishing a comparison with other work;(2)an article might have a higher value for one metric than another paper and a lower value for another metric.

Therefore, the reader cannot deduce from a straightforward comparison of values in metrics. A head-to-head comparison of the results presented in Table 1 would be hasty and unwise as the evaluation metrics were not reported consistently. For instance, Chaudhari et al. [17], Coppock et al. [29], Ponomarchuk et al. [38] and Nguyen et al. [39] used only the AUC metric, while Lella and Pja [40] used accuracy only. Different evaluation metrics are used as follows:(1)$$\mathrm{Accuracy}=\frac{TP+TN}{\mathrm{TP}+\mathrm{TN}+\mathrm{FP}+\mathrm{FN}}$$
(2)$$\mathrm{Precision}=\frac{TP}{TP+FN}$$
(3)$$\mathrm{Specificity}=\frac{\mathrm{TN}}{\mathrm{FP}+\mathrm{TN}}$$
(4)$$\mathrm{Sensitivity}=\frac{\mathrm{TP}}{\mathrm{TP}+\mathrm{FN}}$$
(5)$$F1-\mathrm{score}=2\times \frac{\mathrm{Precision}\times \mathrm{Recall}}{\mathrm{Precision}+\mathrm{Recall}}$$
(6)$$\mathrm{AUC}=\int_{0}^{1}\mathrm{TPR}\left(\mathrm{FPR}\right)d\mathrm{FPR}$$
where FP refers to false positive, FN refers to false negative, TP refers to true positive, and TN refers to true negative.

Other metrics are rarely used and too scarce for their computations to be reported in this review. Among such metrics, the Mathews correlation coefficient considers all four true and false positives and negatives and is thus regarded as a balanced measure in binary classification. In the case of imbalanced data, using the metric UAR (or balanced average) is suggested, while using accuracy is inadvisable or deemed inappropriate. Since a model’s training is conducted using more negative values than positive ones, comparing their numbers during the performance computation provides a skewed ratio. Accuracy is equal for both TPs and TNs divided by all predictions. Therefore, for minimal positive numbers and a greater magnitude of negatives, even though there could be a high rate of FPs, the TNs will dominate. This results in a high accuracy score, poor sensitivity, and possibly high specificity. UAR is the average of sensitivity and specificity, giving an already fairer estimation of a model’s performance. In the same way, the F1 score is defined by the harmonic mean between precision and sensitivity. Additionally, the binary classification model’s essential role is identifying the positive class within the data; naturally, we want to obtain a high rate of correctly predicted positives. Precision and sensitivity help compute the correct overall predicted positives ratio and all elements belonging to the positive class. However, as this task entails a medical diagnosis, predicting an FN is more significant and judged more seriously than an FP. Thus, the FN rate (FNR) might be more appropriate for evaluating one’s model during this task. In addition, the FNR can be straightforwardly computed as FNR=1−Sensitivity. We note that the confusion matrix, a matrix that counts the TN, FN, TP, and FP from actual target and predicted values, is often available, which can provide the reader with all the necessary tools to make a judgment [24,40,41,42].

Overall, the datasets were highly imbalanced: the number of COVID-19-positive samples was much smaller than the negatives and often equaled 1:10. The problem of imbalanced data is consistent in the machine learning field. Having an imbalanced dataset skews a model’s performance evaluation. Among the strategies available to overcome this obstacle, Zhang et al. [43] trained their model after trimming the data. The trained data were filtered to have a perfectly proportioned and “clean” 1:1 the ratio of positives to negatives. The limitation of this curation is that cutting off data will reduce its size, which results in a similar problem. Analogous to data set imbalance, the insufficiency and scantness of a dataset, whether for training or evaluation, is a common problem from which many articles suffer. The reader generally is not provided enough information about the algorithm development to determine if the results are impacted by underfitting or overfitting. Indeed, because of their complexity and layers of abstraction, neural networks are prone to overfitting. Neural networks usually are more data-hungry than classical machine learning algorithms. Hence, data scarcity increases the risk of overfitting occurring. Dataset size also can be the cause of data being too “similar,” meaning that data samples are similar to one another, which also induces a lack of real-world representativeness. It is, for example, the case in Mohammad and Borna [28] who, to overcome this problem, use repeated random split and cross-validation on the Virufy dataset (of 121 samples).

**Table 1 diagnostics-12-02142-t001:** Summary of all included articles on COVID-19 diagnosis from cough using deep learning algorithms.

Author [Year]	Number of Cough Segment and Source of Data	Event Type	Processing	Model	AUC [%]	Sensitivity [%]	Specificity [%]	Precision [%]	Accuracy [%]	F1-Score [%]
Rashid et al. [2022] [44]	15,218 Coughvid	Symptomatic and COVID-19 labels both considered for positive class	Medical information as features passed for training MFCC function of time	CNN	88.9	85.6	85.8	86.3	N/R	86.7
Andreu-Perez et al. [2021] [30]	8380	All PCR tested	MFCC, Mel spectrogram, LPCS into 3D tensor	CNN	98.8	96.43	96.2	96.54	N/R	N/R
Nguyen et al. [2021] [39]	7371 AICovidVN		Feature representations from log-mel spectrograms by EfficientNet-V2	Pre-trained CNN from PANNs	92.8	N/R	N/R	N/R	N/R	N/R
Lella and PJA [2022] [40]	256 Covid-19 Sounds App		Modified MFCC	CNN	N/R	N/R	N/R	N/R	92.32	N/R
Schuller et al. [2020] [45]	1427 Covid-19 Sounds App		Concatenation of raw audio and MFCC	CNN	80.7	69.7	80.2	N/R	73.1	N/R
Ponomarchuk et al. [2021] [38]	23,360 Coughvid, Coswara, and Covid19-Cough;	Partly clinically curated	Mel spectrogram	CNN	80.5	N/R	N/R	N/R	N/R	N/R
	439 clinical data 1395 new for testing		VGGish embedding cochleagram	LightGBM	74.9	N/R	N/R	N/R	N/R	N/R
Lella and Pja [2022] [42]	5000 Covid-19 Sounds App		De-noising Auto Encoder, GFCC, IMFCC, data augmentation	CNN	N/R	N/R	N/R	N/R	93.12	94.13
Chaudhari et al. [2020] [17]	Training 1441 Coughvid, 1442 Coswara Testing 362 Clinical data, 178 Crowdsourced	Training Symptoms, Intensity of symptoms Testing All PCR tested, N/R	-MFCC -Mel spectrogram	Ensemble of -2 separate NN (on MFCC) -CNN (on Mel spectrogram)	72	N/R	N/R	N/R	N/R	N/R
Imran et al. [2020] [36]	543	COVID-19 Bronchitis Pertussis	Mel spectrogram	CNN multiclass classification	N/R	89.14	96.67	89.91	92.64	89.52
			MFCC, PCA projections	SVM multiclass classification	N/R	91.71	95.27	86.6	88.76	89.08
			Mel spectrogram	CNN binary classification	N/R	94.57	91.14	91.43	92.85	92.97
Akinnuwesi et al. [2021] [46]	600 patient records	Clinical data	Random over and under-sampling	MLP	N/R	86.3	N/R	N/R	88.3	86.9
				FCM	N/R	79.5	N/R	N/R	79.2	82.3
Jayachitra et al. [2021] [41]	289 Virufy, Coswara		MFCC	CNN	N/R	94.11	N/R	96.96	97.12	96
Soltanian and Borna [2022] [28]	121 Virufy	All PCR tested	MFCC	CNN	N/R	90	100	100	95	94.7
				Two Separable Quadratic Convolutional Layers (inspired by LeNet-1)	N/R	95.2	100	100	97.5	97.6
Mohammed et al. [2021] [32]	1502 Coswara		Spectrogram, mel spectrum, raw data, MFCC, power spectrum, chromatogram	CNN ensemble	77	71	N/R	80	N/R	75
Lella and PJA [2021] [47]	1539 Covid-19 Sounds App		Data augmentation, DAE, IMFCC, GFCC	CNN	N/R	N/R	N/R	N/R	95.45	96.96
Tawfik et al. [2022] [48]	1171 Coswara 121 Virufy		Chroma, ZCR, QCT, MFCC	CNN	N/R	99.6	99.7	N/R	98.5	98.4
Zhang et al. [2021] [43]	321 hospitalized positive patients NR from ESC50, DCASE2016, Virufy, Coswara and Coughvid		MFCC	CNN	98.1	95.3	95.8	97.0	95.8	96.1
Dentamaro et al. [2022] [24]	49 Covid Sounds App	Cough symptomatic healthy control class		4 SE-ResNet blocks and 4 CBAM blocks trained on UrbanSound 8K	92.57	80.39	N/R	88.81	92.56	89.8
Dentamaro et al. [2022] [24]	1118 added with Coswara			4 SE-ResNet blocks and 4 CBAM blocks trained on UrbanSound 8K	81.86	76.88	N/R	73.26	76.88	70.98
Pahar et al. [2022] [23]	1171 Coswara, 44 Sarcos 517 Compare, 11202 TASK, Google Audio Set, etc.		MFCC, filterbank energies, kurtosis, ZRC, Voluntary coughs dataset fed for pre-training and transfer learning	CNN	97.2	98	92	N/R	95	N/R
				LTSM	96.4	95	93	N/R	94	N/R
				ResNet-50	98.2	98	97	N/R	97	
Sabet et al. [2022] [49]	N/R Coughvid		Data augmentation, MFCC	CNN	94	94	N/R	93	93	92
Loey and Mirjalili [2021] [26]	1457 Coughvid		Scalogram	ResNet-18	N/R	94.44	95.37	95.33	94.9	94.88
				ResNet-50	N/R	92.59	88.89	89.29	N/R	90.91
				ResNet-101	N/R	97.22	86.11	87.5	N/R	92.11
				MobileNet	N/R	91.67	86.11	86.84	N/R	89.19
				NasNet	N/R	89.81	89.81	89.81	N/R	89.81
				GoogleNet	N/R	93.52	87.96	88.6	N/R	90.99
Fakhry et al. [2021] [31]	5749 Coughvid	Asymptomatic COVID-19 negative, Symptomatic, COVID-19 negative COVID-19 positive	Gaussian noise data augmentation, pitch shifting, MFCC	Multi-branch ensemble ResNet-50 DNN SFFN	99	85	99.2	N/R	N/R	N/R
			Spectrogram	ResNet-50	97	64	97.1	N/R	N/R	N/R
Laguarta et al. [2020] [50]	4256 MIT Open Voice		MFCC	3 pre-trained parallel ResNet-50	97	98.5	94.2	N/R	98.5	N/R
Coppock et al. [2021] [29]	517 (from 355 patients) Covid-19 Sounds App		Spectrogram extraction	ResNet	84.6	N/R	N/R	N/R	N/R	N/R
				SVM	72.1	N/R	N/R	N/R	N/R	N/R
Pahar et al. [2021] [10]	1171 Coswara 44 Sarcos	Clinically tested	MFCCs, log frame energy, ZCR and kurtosis	ResNet-50	74.2	93	57	N/R	74.58	N/R
				LSTM and features sequential forward selection	93.8	91	96	N/R	92.91	N/R
Bhanusree et al. [2022] [51]	549 Covid-19 Sounds App		Data augmentation with Gaussian noise	Architecture of CNN1d and bidirectional LSTM layers	N/R	77.93	96.18	97.16	86	86.49
Al-Dhlan [2021] [52]	N/R		Noise reduction using LMS MFCC	GAN classifier	N/R	96.15	N/R	96.54	98.56	97
Hamdi et al. [2022] [53]	16,082 Coughvid		Mel spectogram, pitch-shifting, spectral data augmentation	Attention-based hybrid CNN-LTSM	89.28	87.74	90.81	89.46	89.35	88.56
Hassan et al. [2020] [54]	80 Hospitals	20 Hospitalised Covid-19 patients	MFCC	LSTM	97.4	96.4	N/R	99.3	97	97.9
Khriji et al. [2021] [55]	6593 ESC-50, AudioSet	Datasets annotated with BMAT	MFCC	LSTM	N/R	78.75	N/R	79	80.26	79
Feng et al. [2021] [56]	200 Coswara 16 Virufy	Virufy clinically curated	ZRC, energy, entropy; spectral centroid, spectral spread, entropy, MFCCs	SimpleRNN	92.82	N/R	N/R	N/R	90	N/R
Han et al. [2022] [34]	5240 (2478 patients) Covid-19 Sounds App	Recent tests	VGGish feature extraction	CNN	66	59	66	N/R	N/R	N/R
Xia et al. [2021] [37]	38,869 Covid-19 Sounds app		Filter out noisy data with YAMNet	Pre-trained VGGish	62	69	45	N/R	N/R	N/R
				VGGish	66	59	66	N/R	N/R	N/R
Chowdhury et al. [2021] [35]	1039 Covid-19 Sounds App, Nococoda, Coswara, Virufy	Asymptomatic and symptomatic	Mel frequency, cepstral coefficients, mel spectrogram, chromagram, spectral contrast	AdaBoost	80	55	94	82	84	66
				MLP	83	51	95	84	81	63
				HGBoost	83	56	98	92	84	70
Islam et al. [2022] [57]	100 Virufy		Spectral centroid, spectral entropy, spectral flux, spectral roll-offs, MFCC, and chroma vector	DNN	N/R	95	N/R	100	97.5	97.4
Bagad et al. [2020] [58]	3117 CoughDetect app	376 PCR tested positive	Mel spectrogram	ResNet-18 (ImageNet)	68	N/R	31	N/R	N/R	N/R

Moreover, an imbalance in the data can significantly impact the overall performance of models. There are not enough studies in the literature where COVID-19 classification models are tested against possible biases. Han et al. [34] determined that it was a worthy measure to ensure that their model is indeed resilient and does not confuse ailments and other health issues with COVID-19. Another gap in the literature is the absence of studies testing COVID-19 detection models against other types of coughs, without which there is no concrete proof that a neural network indeed detects COVID-19 or another respiratory illness from a patient’s cough. Fakhry et al. [31] contrasted different age intervals and genders in their study. In comparing gender, the performance metrics were recorded as higher for the female group: 81.12% vs. 74.4% for sensitivity and 85% vs. 77% for AUC. In comparing age, the highest sensitivity, specificity, and AUC were attained for the coughs of patients under 20 years old (85%, 100%, and 92%, respectively). The performance of the model on the group of patients aged 20–40 years old was 73% for AUC, 75.8% for sensitivity, and 97.9% for specificity. The model’s performance on the group of patients aged 40–60 years old was 91% for AUC, 72.7% for sensitivity, and 100% for specificity. The performance of the model on the group of patients older than 60 years was 50% for AUC, 75% for sensitivity, and 100% for specificity. Bias could occur if the dataset is unbalanced regarding age, gender, and sample size. Therefore, Imran et al. [36] introduced a third prediction category for their algorithm called “inconclusive”.

Going through the included articles, we can see that the ground truth to which the performance of models is computed is not exact. Moreover, almost all articles’ datasets used during training neural networks are prone to label noise. First, they are crowdsourced, which is a source of the unreliability of data. Indeed, crowdsourcing leads to inconsistencies and inaccuracies in labels and other metadata information extracted from data. Furthermore, we note that even PCR tests still present a risk for FPs.

Evaluating a model’s capacity and performance requires a dataset with considerable sample size and different types of acoustics. To answer the question, “Does the algorithm work?” one must test an algorithm on a large and diversified dataset. For example, one should try their model outputs in cases where there is no cough present in the audio file at all or with an increasing amount of noise, among others, and determine to what extent the machine predicts the truth, bearing in mind that the objective is ultimately to build an ideal model for real-world healthcare applications.

An idea for testing a deep learning model against unfairness is to train the said model with all the subgroups separated by age, gender, ethnicity, language, COVID-19 duration, COVID-19 recovery, intensity status, other illnesses, whether the cough is symptomatic or voluntary, frequency bandwidths, and quality of audio. Finally, some matters remain unanswered. Are neural networks capable of overcoming all biases? What metrics ought to be used to evaluate their performance objectively? How does data fidelity impact the analysis? For instance, the impact of sampling rate is still uncertain: Nguyen et al. [39] model’s scores were 95%, 97% and 99% for a sampling rate set to 4 Hz, 8 kHz, and 48 kHz, respectively. Is this due to the loss of data and overfitting? We can sum up a list of recommendations for future research to address the identified limitations aforementioned:Collect audio signals from a large population (n>1000) of clinically tested for both negative and positive COVID-19.Include cough signals from a more diverse set of respiratory diseases, including different variants of the coronavirus.Develop a balanced dataset with a diverse population of age, gender, and ethnicity.Evaluate developed algorithm using the follow metrics: specificity, sensitivity, accuracy, AUC, MCC and F1-score.

## 5. Conclusions

We conclude that cough sounds can be used as biomarkers for identifying coronavirus and respiratory diseases and can be complemented by breath and speech sounds to improve detection accuracy. We also highlighted current limitations in the existing methods and recommended a few steps to move this technology forward. Modern healthcare tools, such as automated diagnostics devices, have drawn more focus thanks to the increased computational power and medical data. There still are different barriers to accumulating large clinical data, notably patient privacy concerns. However, datasets’ quality and size are crucial to healthcare services’ success. Moreover, in machine learning, transfer learning and the availability of large public datasets significantly increase neural networks’ performances. Nonetheless, further research with more diverse datasets, especially regarding respiratory diseases, is warranted to test neural network architectures against all forms of biases.

## Figures and Tables

**Figure 1 diagnostics-12-02142-f001:**
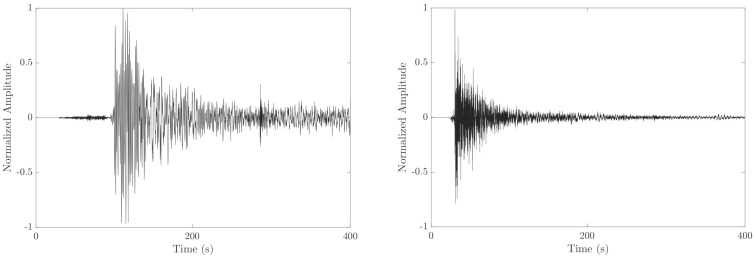
Raw data of cough signals extracted from the crowdsourced publicly available dataset Coughvid [12]. The dataset’s recordings were collected from participants using their built-in computer microphone, the sampling rate of the data is 48 kHz. (**Left panel**) COVID-19-positive: symptomatic 40-year-old male; (**Right panel**) COVID-19-negative: healthy 46-year-old male.

**Figure 2 diagnostics-12-02142-f002:**
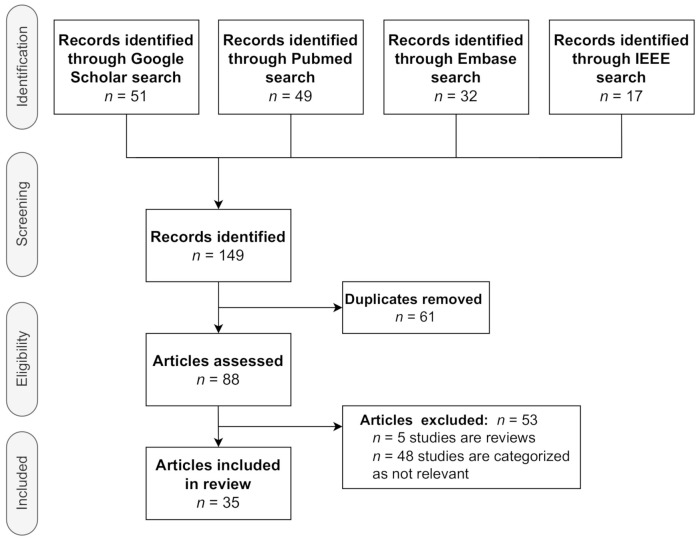
Literature search.

**Figure 3 diagnostics-12-02142-f003:**
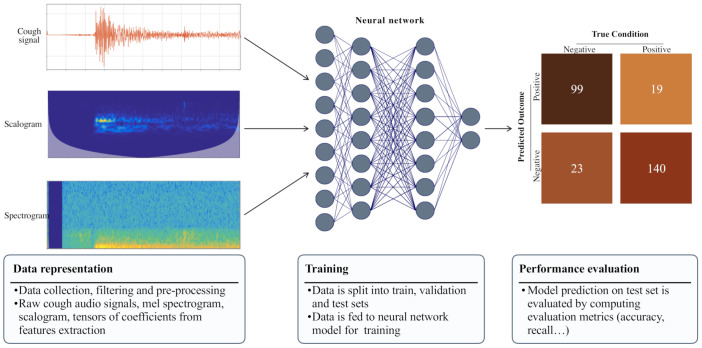
Pipeline of the general experimental procedures.

**Figure 4 diagnostics-12-02142-f004:**
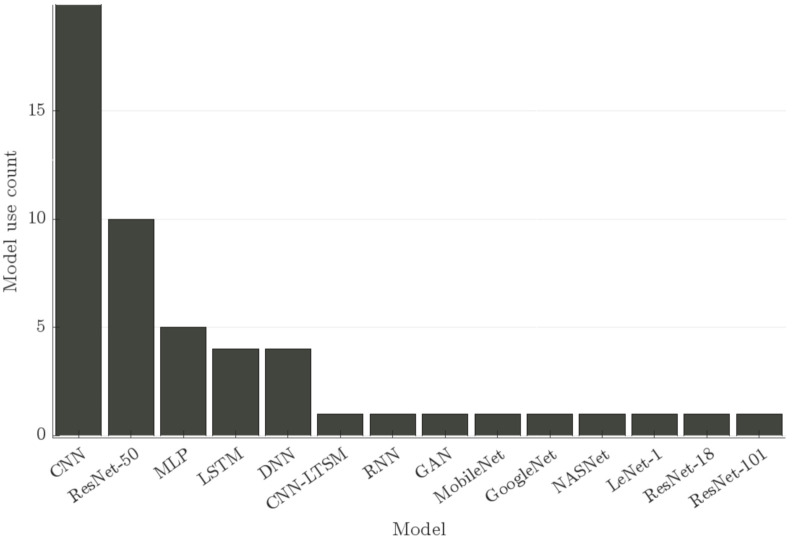
Count of the deep learning models.

## Data Availability

Not applicable.

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
