# Peer review of "Challenges and Opportunities of Deep Learning for Cough-Based COVID-19 Diagnosis: A Scoping Review"

_diagnostics, 2022, doi:10.3390/diagnostics12092142_

Round 1
Reviewer 1 Report
The paper provides a deep analysis of the state of research on the development of neural network models in the context of the diagnosis of COVID-19.
The paper is well organized and describes a good methodology and an interesting research investigation field.
I would suggest the authors to improve the paper and the analysis, for example to add more details in the discussion section that can be enriched of details and of analysis.
The metrics for example are introduced but not well described and contextualized in the analysis. When are these metrics used? Are they used
in the same way? How much the data size is influencing a good usage of the metrics? The authors say just before that the metrics are used in just one paper
but if it is true then why are they presented? Is this one-single-usage enough to put these metrics such in advance?
The conclusions are foggy. I suggest the authors to better motivate the conclusions (ie, the sentence "The challenges encountered by neural networks in detecting coughs have also been elucidated.")
Author Response
1. I would suggest the authors to improve the paper and the analysis, for example to add more details in the
discussion section that can be enriched of details and of analysis. The metrics for example are introduced
but not well described and contextualized in the analysis.
When are these metrics used?
Are they used in the same way?
How much the data size is influencing a good usage of the metrics?
The authors say just before that the metrics are used in just one paper but if it is true then why are they
presented?
Is this one-single-usage enough to put these metrics such in advance?
Author response: The author agrees that there is confusion regarding the metrics subsection.
Author action: The author expanded on the metrics section of the discussion. The link between the metrics and the table was made clearer (e.g., by referencing the table). The author has also elaborated on the discussion, introduced a list of future recommendations for research, and added clearer explanations and logical transitions to each idea. The author has also cleaned and reduced the size of the table to make it more readable. The whole manuscript was checked and now it reads more clear after addressing this point.
2. The conclusions are foggy. I suggest the authors to better motivate the conclusions (ie, the sentence "The
challenges encountered by neural networks in detecting coughs have also been elucidated.")
Author response: The author agrees with this statement.
Author action: The conclusion was rewritten, introducing clear action verbs and logical order. The sentence "The challenges encountered by neural networks in detecting coughs have also been elucidated." was removed.
Reviewer 2 Report
This Paper is better defined and the Parameter Setting is good.
But the less elaborate conclusion, needs for widely discussed in the conclusion.
Author Response
This Paper is better defined and the Parameter Setting is good.
Author response: We thank the reviewer for the positive feedback.
Author action: None.
But the less elaborate conclusion, needs for widely discussed in the conclusion.
Author response: We thank the reviewer for the valuable suggestion.
Author action: We modified the discussion and add a recommendation list. The whole manuscript is checked and many parts were rewritten.
Reviewer 3 Report
The manuscript should be further improved.
The English writing is not professional enough and there are many grammar or syntax mistakes.
Analysis is not solid. The authors failed to compare their work with newly published literatures, which makes it hard to evaluate the innovations of the proposal in this paper.
In the introduction part of this paper, the author simply enumerates the recent literature, and does not give the relevant thinking results.
The novelty and contributions of this work are limited. Only authors have used various technologies without highlighting their integration issues
How the pre-processing of the data carried out?
How the loss function is formulated and its significance?
The motivation of this paper is not well described and readers may not understand why the authors want to work on this method.
The quality of this work can not be accepted for publication in such a high quality journal. Besides, the writing is somewhat poor. More analysis should be added.
There are a number of parameters including experimental parameters seem mostly ad-hoc and empirical, as listed from Equations 1 to 6. Too many parameters may cause uncertainty to the efficacy evaluation of the proposed work. Moreover, how these parameters influence the experimental results, is not clear and should be further evaluated.
Anaysis is not sufficient. More comparison with state-of-the-art and datasets are needed. More simulation results are needed to validate the effectiveness of the proposed scheme.
The comparison seems somewhat unsuitable. The manuscript requires careful proofreading, especially the format of the references There are many typos and grammar errors.
Finally, the conclusions are very superfluous and seem more than conclusions a summary of the introduction. It is important to improve the conclusions based on the results obtained.
Author Response
1. The manuscript should be further improved. The English writing is not professional enough, and there are many grammar or syntax mistakes.
Author response: The author agrees with this statement.
Author action: The manuscript was proofread.
2. The authors failed to compare their work with newly published literatures, which makes it hard to evaluate the innovations of the proposal in this paper.
Author response: There is a misunderstanding; this paper is a literature review paper, not a methodological article. The authors did not introduce a new method to detect COVID.
3. In the introduction part of this paper, the author simply enumerates the recent literature, and does not give the relevant thinking results. The novelty and contributions of this work are limited. Only authors have used various technologies without highlighting their integration issues. How the pre-processing of the data carried out? How the loss function is formulated and its significance? The motivation of this paper is
not well described and readers may not understand why the authors want to work on this method.
Author response: There is a misunderstanding; this paper is a literature review paper, not a methodological article. The authors did not introduce a new method to detect COVID.
5. Anaysis is not sufficient. More comparison with state-of-the-art and datasets are needed. More simulation
results are needed to validate the effectiveness of the proposed scheme.The comparison seems somewhat unsuitable.
Author response: There is a misunderstanding; this paper is a literature review paper, not a methodological article. The authors did not introduce a new method to detect COVID.
4. There are a number of parameters including experimental parameters seem mostly ad-hoc and empirical, as listed from Equations 1 to 6. Too many parameters may cause uncertainty to the efficacy evaluation of the proposed work. Moreover, how these parameters influence the experimental results, is not clear and should be further evaluated.
Author response: The author did not introduce his own work.
Author action: The author expanded on the metrics section of the discussion to clarify the purpose and relevance of the metrics definitions (equations 1 to 6) . The author expanded on the metrics section of the discussion. The link between the evaluation metrics and the table was made clearer with references and examples. The table is now modified, and it's clearer. We elaborated on the discussion overall, introduced a list of future recommendations for research, and added clearer explanations and logical transitions to the ideas of the discussion.
Author response: The author did not introduce his own work.
6. The manuscript requires careful proofreading, especially the format of the references. There are many
typos and grammar errors.
Author response: The author agrees with this statement.
Author action: The manuscript was proofread.
7. Finally, the conclusions are very superfluous and seem more than conclusions a summary of the
introduction. It is important to improve the conclusions based on the results obtained.
Author response: The author did not introduce his own results. Nonetheless agrees on the confusion that the
conclusion may lead in the minds of the readers.
Author action: The conclusion was rewritten introducing clear action verbs and logical order. The sentence "The
challenges encountered by neural networks in detecting coughs have also been elucidated." was removed.
Round 2
Reviewer 1 Report
The paper provides a deep analysis of the state of research on the development of neural network models in the context of the diagnosis of COVID-19.
Reviewer 3 Report
The revised version can be accepted for publication